# Metatranscriptomic Analysis of Human Lung Metagenomes from Patients with Lung Cancer

**DOI:** 10.3390/genes12091458

**Published:** 2021-09-21

**Authors:** Ya-Sian Chang, Ming-Hung Hsu, Siang-Jyun Tu, Ju-Chen Yen, Ya-Ting Lee, Hsin-Yuan Fang, Jan-Gowth Chang

**Affiliations:** 1Epigenome Research Center, China Medical University Hospital, Taichung 404, Taiwan; t25074@mail.cmuh.org.tw (Y.-S.C.); t24399@mail.cmuh.org.tw (J.-C.Y.); t23701@mail.cmuh.org.tw (Y.-T.L.); 2Center for Precision Medicine, China Medical University Hospital, Taichung 404, Taiwan; t35492@mail.cmuh.org.tw (M.-H.H.); t34752@mail.cmuh.org.tw (S.-J.T.); 3School of Medicine, China Medical University, Taichung 404, Taiwan; 4Department of Thoracic Surgery, China Medical University Hospital, Taichung 404, Taiwan; d17573@mail.cmuh.org.tw; 5Department of Bioinformatics and Medical Engineering, Asia University, Taichung 413, Taiwan

**Keywords:** metatranscriptomic, lung cancer, microenvironments, T cell receptor repertoires, B cell receptor repertoires

## Abstract

This study was designed to characterize the microbiomes of the lung tissues of lung cancer patients. RNA-sequencing was performed on lung tumor samples from 49 patients with lung cancer. Metatranscriptomics data were analyzed using SAMSA2 and Kraken2 software. 16S rRNA sequencing was also performed. The heterogeneous cellular landscape and immune repertoires of the lung samples were examined using xCell and TRUST4, respectively. We found that nine bacteria were significantly enriched in the lung tissues of cancer patients, and associated with reduced overall survival (OS). We also found that subjects with mutations in the epidermal growth factor receptor gene were less likely to experience the presence of *Pseudomonas. aeruginosa*. We found that the presence of CD8+ T-cells, CD4+ naive T-cells, dendritic cells, and CD4+ central memory T cells were associated with a good prognosis, while the presence of pro B-cells was associated with a poor prognosis. Furthermore, high clone numbers were associated with a high ImmuneScore for all immune receptor repertoires. Clone numbers and diversity were significantly higher in unpresented subjects compared to presented subjects. Our results provide insight into the microbiota of human lung cancer, and how its composition is linked to the tumor immune microenvironment, immune receptor repertoires, and OS.

## 1. Introduction

Lung cancer is the most common cancers worldwide, and its incidence is rapidly increasing in Taiwan. Globally, there are 1.8 million lung cancer diagnoses and 1.6 million deaths per year [1]. Chemical carcinogens, chronic inflammation, bacterial and viral infections, periodontal diseases, and various other factors promote lung cancer development. Several pathogenic microorganisms are associated with lung cancer including *Haemophilus influenzae*, *Acidovorax*, *Klebsiella*, *Moraxella catarrhalis*, *Mycobacterium tuberculosis* and *Granulicatella adiacens* [2]. Despite recent advances in targeted receptor therapies and immunotherapies, the five-year survival rate of lung cancer remains low. Major reasons for the poor prognosis include late diagnosis and resistance to standard chemotherapy [3].

Over the last decade, a number of studies have investigated the microbiome of cancer patients, including patients with oral [4], lung [5], stomach, esophageal [6], pancreatic [7], intestinal [8], and prostate [9] cancers. Straussman et al. performed 16S rRNA sequencing on 1526 samples collected from patients at nine medical centers with seven cancer types; they could distinguish between various cancer types based on the microbial DNA present in the samples [10]. In the same year, Knight et al. analyzed the microbial reads in whole-genome and whole-transcriptome sequencing data from The Cancer Genome Atlas for 33 cancer types, and found associations between different cancer types and specific microbiota [11].

The first study of lung tissue microbiota was conducted in 2016 [12], and revealed that the genus *Thermus* is significantly more abundant in advanced-stage cancer patients (stage IIIB and IV). Furthermore, *Acidovorax*, *Cyanobacteria*, *Streptococcus* and *Prevotella* are enriched in lung cancer tissues [5,13,14]. Another recent study explored the relationship between the microbiome of resected lung tissue and lung cancer prognosis [15], and revealed that greater diversity of normal tissue was associated with poorer recurrence-free and disease-free survival. Recently, the lung microbiota of bronchoalveolar lavage fluid and non-malignant, peritumoral and tumor tissue samples, from 18 non-small cell lung cancer (NSCLC) patients, ref. [16] were examined; bronchoalveolar lavage fluid was found to harbor unique microbiota, but few differences were found among the other tissues. In the three types of lung tissue samples, *Pseudomonas*, *Clostridium*, *Kocuria*, *Acinetobacter,* and *Sphingomonas* were the five most abundant genera. All of the studies discussed above used 16S rRNA sequencing to examine human lung tissue microbiota. However, metatranscriptomic analysis has not yet been performed.

Advances in next generation sequencing technologies, including 16S rRNA sequencing and metagenomic technologies, have enabled the study of the microbial gene repertoire. RNA-sequencing (RNA-seq) has facilitated the detection of microorganisms in a diverse range of microbiomes. Moreover, metatranscriptomics has been applied to study various environments, including human hosts, plants, soils, and aquatic environments [17]. Researchers have applied metatranscriptomics to investigate the interactions between microbes and human host [18], and to explore associations of microbes with disease progression [19] and severity [20]. However, this technique has not yet been applied to human lung cancer tissues.

In this study, we conducted a metatranscriptomics pilot study using cancer tissues collected from 49 patients with NSCLC. Eight samples were paired tumor and non-disease samples. Using RNA-seq, we explored the associations between the tumor metatranscriptome, tumor immune microenvironment, and immune receptor repertoire.

## 2. Materials and Methods 

### 2.1. Sample Population

Tissue specimens were obtained from 49 Taiwanese patients with lung cancer who underwent surgical resection between May 2007 and April 2014 at China Medical University Hospital. Forty-nine lung tumors were examined, including forty adenocarcinomas and nine squamous cell carcinomas. Surgically resected specimens were grossly dissected and preserved immediately in liquid nitrogen following surgery. The present study was approved by the Institutional Review Board of the China Medical University Hospital (CMUH106-REC1-053).

### 2.2. RNA-Seq

Total RNA was extracted from the clinical tissue samples using a NucleoSpin^®^ RNA Kit (Macherey-Nagel GmbH, Düren, Germany) according to the manufacturer’s instructions. The quality, quantity, and integrity of the extracted RNA were evaluated using a NanoDrop1000 spectrophotometer and an Agilent 2100 Bioanalyzer (Agilent Technologies, Santa Clara, CA, USA). Samples with RNA integrity >6.0 were used for RNA-seq. An mRNA-focused, barcoded library was generated using a TruSeq Stranded mRNA Library Preparation Kit (Illumina, San Diego, CA, USA). The libraries were sequenced using the Illumina Nova Seq 6000 platform (Illumina), using 2 × 151 bp paired-end sequencing flow cells according to the manufacturer’s instructions.

### 2.3. RNA-Seq Data Analysis

The RNA-seq data were analyzed as described previously [21]. In brief, data quality control at the Q20 level was performed using Trimmomatic [22], read alignment to the GRCh38 human genome was conducted using HISAT2 [23], expression was quantified using GENCODE v22 (excluding including mitochondrial genes), and transcripts were normalized into transcripts per million (TPM) using StringTie [24].

### 2.4. Cell Enrichment Analysis

xCell [25] was used to examine the enrichment of various immune cells in the tumors, and to calculate an immune score from the TPM expression matrix.

### 2.5. Detection and Analysis of Immune Receptor Repertoires

To characterize the immune receptor repertoires from the RNA-seq data, we used TRUST4 (v1.0.2) [26] software, which was applied to download the IMGT Repertoire reference [27], perform *de novo* assembly, and annotate and count consensus assemblies of T cell receptors (TCRs) and B cell receptor (BCRs). The clonal diversity of TCRs and BCRs was calculated using VDJtools (v1.2.1) [28].

### 2.6. Metatranscriptomic Analysis

SAMSA2 [29] and Kraken2 [30] were used independently to analyze metatranscriptomic data. The first step in the SAMSA2 pipeline (v2.2.0) was to merge paired-end reads using PEAR (v0.9.6) [31], and then to remove bacteria rRNA reads using SortMeRNA (v2.1) [32]. Then, DIAMOND (v0.9.36) was used to annotate the transcriptome using the National Center for Biotechnology Information (NCBI) [33] Reference Sequence (RefSeq) database [34]. Sequences with >96% similarity were selected for annotation. Kraken2 (--kmer-len = 120, v2.1.1) [30] was also applied to analyze the metatranscriptomic data after removing rRNA reads. The NCBI RefSeq database was used to annotate transcripts against human, virus, archaea, bacterial and fungal genomes (database version: July, 2020).

### 2.7. 16S rRNA Sequencing

DNA was extracted from the clinical tissue samples using a DNeasy Blood and Tissue Mini kit (Qiagen, Valencia, CA, USA) according to the manufacturer’s instructions. Extracted DNA was quantified using Qubit (Life Technologies, Grand Island, NY, USA). The V4 region of the 16S rRNA gene was PCR-amplified using the 515F/806R primer pair [35], which contain Nextera adapter sequences (Illumina) at their 5′-ends, for library preparation (515F: 5′-GTGCCAGCMGCCGCGGTAA-3′ and 806R: 5′-GGACTACHVGGGTWTCTAAT-3′). PCR was performed using a KAPA HiFi HotStart ReadyMix PCR Kit (Roche, Cape Town, South Africa). The PCR program consisted of 30 s at 98 °C followed by 30 cycles of 10 s at 98 °C, 30 s at 60 °C, and 30 s at 72 °C, with a final amplification step of 5 min at 72 °C. PCR products were purified using the AMPure XP Beads (Beckman Coulter, Indianapolis, IN, USA) and quantified using an Agilent 2100 Bioanalyzer (Agilent Technologies, Santa Clara, CA, USA). A Second round of PCR using Illumina dual-index oligos was performed using the KAPA HiFi HotStart ReadyMix PCR Kit (Roche, Basel, Switzerland), as follows: 95 °C for 3 min, eight cycles of 95 °C for 30 s, 55 °C for 30 s and 72 °C for 30 s, and a final amplification step at 72 °C for 5 min. Samples were pooled and purified using AMPure XP Beads (Beckman Coulter, Brea, CA, USA). Sequencing was performed on the Illumina Miseq instrument using the MiSeq reagent kit v3 (600 cycles).

### 2.8. Bioinformatic Analysis of 16S rRNA Sequencing Data

Sequencing data were analyzed using Illumina local run manager software. In brief, the index reads were demultiplexed, FASTQ files were generated, and the reads were classified against the Greengenes 16S rRNA gene database (version gg_13_5) [36], which achieved up to species-level sensitivity.

### 2.9. Statistical Analysis

Differences between groups were compared using the Mann-Whitney test. The log-rank test and Cox proportional hazards regression model was used to compare differences in overall survival (OS) between groups. Relationships between bacterial infection status and gene mutation status were determined using Chi-square test or Fisher’s exact test. All statistical analyses were calculated using SciPy (v1.2.1) package [37], lifelines (v0.22.3) package [38], GraphPad Prism 8.0.2, or SPSS 22.0. A *p*-value of less than 0.05 was considered statistically significant.

## 3. Results

### 3.1. Microorganisms Identified in Lung Cancer

A bacterial infection was considered to be present if the bacterial species was detected by both the SAMSA2 and Kraken2 tools, the bacterium was detected by SAMSA2 and 16S rRNA sequencing, or the bacterium was detected by Kraken2 and 16S rRNA sequencing. Using these criteria, 435 taxonomic groups were detected (Appendix A). We applied a log-rank test of OS to evaluate the prognostic impact of each bacterial species, and found that 63 taxonomic groups were associated with survival in lung cancer patients (Appendix A). We subjected *Brevundimonas diminuta* (*n* = 3), *Acinetobacter radioresistens* (*n* = 5), *Enterobacter cloacae* (*n* = 3), *Mycobacterium chelonae* (*n* = 3), *Mycobacterium franklinii* (*n* = 5) and *Staphylococcus sp.* (*n* = 3) to further analysis (Figure 1). 

Moreover, we conducted survival analysis of patients with bacterial presences detected by SAMSA2 and Kraken2 analyses of RNA-seq and 16S rRNA sequencing data. Fifty taxonomic groups were detected, of which seven were associated with survival in lung cancer patients (Appendix A). We subjected *Bacillus megaterium* (*n* = 2), *P. aeruginosa* (*n* = 4) and *Rhodococcus erythropolis* (*n* = 2) to further analysis (Figure 1).

A virus, human papillomavirus (HPV) type 16, was identified in the metatranscriptome using SAMSA2 and Kraken2 (Figure 1). However, HPV type 16 was not associated with the OS of lung cancer patients.

### 3.2. Cell Types Associated with Survival in NSCLC

Figure 2 shows the cellular heterogeneity of each sample based on the RNA-seq data. We see that xCell estimates of immune and stromal cell types can be used to cluster two different lung cancer subtypes. Of the 64 cell types identified, 34 were immune cells. We found that five immune cell types were significantly associated with patient survival: CD8+ T-cells, CD4+ naive T-cells, dendritic cells (DC) and CD4+ central memory T-cell (CD4+ Tcm) were associated with high OS (Figure 3A), while pro B-cells enrichment was predictive of adverse survival outcomes (Figure 3B). Appendix A showed the hazard ratios of the impacts of tumor immune cells on lung cancer. The CD8+ T-cells, CD4+ naive T-cells, DC and CD4+ Tcm maintained the significance after adjustment for stage, grade, gender and histologic cell type (*p* = 0.004, 0.046, 0.04 and 0.035, respectively) (Appendix A). After adjustment, the pro B-cells showed borderline significance (*p* = 0.058). 

### 3.3. Compositional Changes in Tumor Immune Cell Populations

We identified 13 immune cell subtypes, and ImmuneScore and MicroenvironmentScore differed significantly between presented and unpresented subjects (Appendix A). In the multivariate analysis which incorporated independence prognostic factors of 13 immune cell subtypes, ImmuneScore and MicroenvironmentScore, we found that there is a significant association between presented subjects and poor survival (*p* = 0.001) (Figure 4).

*B. diminuta* presence was associated with CD4+ T-cells and regulatory T-cells (Tregs) enrichment (Appendix A). *A. radioresistens* presence was associated with relatively low level of activated DC (aDC), CD4+ effector memory T-cells (CD4+ Tem), CD8+ central memory T-cells (CD8+ Tcm), macrophages, macrophages M2, memory B-cells and plasma cells, as well as a low ImmuneScore and MicroenvironmentScore (Appendix A). *E. cloacae* presence was linked to relatively high levels of macrophages M1 and Tregs, and relatively low levels of CD4+ naive T-cells (Appendix A). Presence with *M. chelonae* and *M. franklinii* was associated with relatively high levels of CD4+ T-cells (Appendix A). *Staphylococcus sp.* presence correlated with relatively high levels of basophils, eosinophils, naive B-cells, natural killer (NK) cells, and pro B-cells (Appendix A).

*B. megaterium* presence was associated with relatively high levels of type 2 T-helper (Th2) cells and relatively low levels of aDC, natural killer T (NKT) cells and plasma cells (Appendix A). *R. erythropolis* presence was linked to relatively high levels of CD4+ T-cells (Appendix A). *P. aeruginosa* and HPV type 16 presences were not significantly associated with the enrichment of any specific tumor immune cells.

We further analyzed whether the positive effect of presence of bacterial on the survival is related to their positive effect on levels of tumor immune cells or it is an independent effect. The *B. diminuta*, *M. chelonae* and *M. franklinii* maintained the significance after adjustment for tumor immune cells (*p* = 0.031, 0.006 and 0.031, respectively) (Appendix A). After adjustment, the *A. radioresistens*, *E. cloacae*, *Staphylococcus sp.*, *B. megaterium* and *R. erythropolis* showed no significance of the relationship.

### 3.4. TCR and BCR Repertoires in NSCLC

We systematically analyzed TCR and BCR repertoires in RNA-seq data from 48 lung cancer tissues. One case has not passed the TRUST4 criteria. Detailed information about the TCR (α, β, γ and δ) and BCR (IgL, IgK and IgH) repertoires is included in Appendix A. Multivariate analysis, which included tumor stage, grade, histologic cell type and gender as independent prognostic factors, revealed that the clone numbers of TCRα, significantly correlated with OS (*p* = 0.044) (Table 1). For all TCR and BCR repertoires, the ImmuneScore was significantly higher in groups with higher clone numbers (Figure 5A–G).

### 3.5. Compositional Differences between TCR and BCR Repertoires

Next, we analyzed the relationships of bacterial presence with the number of unique immune receptor clones and clonal diversity in 48 subjects. We found that patients with bacterial presences had fewer unique TCR and BCR clones compared to unpresented subjects (Figure 6A). In addition, the clonal diversity of T and B cells was significantly higher in unpresented subjects (Figure 6B). However, we did not find any significant associations between HPV type 16 presence and immune receptor repertoires.

### 3.6. Associations between the Metatranscriptome and Genetic Alterations

The most prevalent mutations in lung tumors are found in the epidermal growth factor receptor (*EGFR*), *KRAS*, phosphatidylinositol-4,5-bisphosphate 3-kinase catalytic subunit α and *TP53* genes. Therefore, we investigated the association between the lung metatranscriptome and host genetic mutations. We found a higher rate of mutation in the *EGFR* gene among patients not presented with *P. aeruginosa*, exhibiting borderline statistical significance (*p* = 0.050) (Appendix A). However, we found no other associations between bacterial species and gene mutations.

## 4. Discussion

This is the first study to employ RNA-seq to investigate the metatranscriptome of human lung cancer. *B. diminuta*, *A. radioresistens*, *E. cloacae*, *M. chelonae*, *M. franklinii*, *Staphylococcus sp.*, *B. megaterium*, *P. aeruginosa,* and *R. erythropolis* were enriched in lung cancer tissues and significantly associated with prognosis. Most of the bacterial species identified in this study (except for *M. franklinii* and *B. megaterium*) were consistent with those detected in a recent study, in which 16S rRNA sequencing was applied to identify species present in lung tissues [10]. Using xCell, we estimated the relative proportions of immune cells, and investigated their relationships with the metatranscriptome. We also investigated immune receptor repertoires, and their associations with species identified in the metatranscriptome. Moreover, we found that *EGFR* mutations may protect against *P. aeruginosa* presences.

*Bacillus sp.* produce anti-cancer and anti-proliferative biomolecules [39]. *B. megaterium* forms large spores, which distinguish it from other *Bacillus sp*. Moreover, *B. megaterium* strain SAmt17 produces extracellular polymeric substances that suppress the expression of hepatocyte carcinoma G2 cells, and strain ATCC 13368 produces four cytotoxic compounds suppressing human melanoma cells. Infections caused by *B. megaterium* are rare; only five cases (eye, skin, brain, pleuritis with pleural effusion [40], and soft tissue infections [41]) have been described in the literature so far. In the present study, we detected *B. megaterium* in two lung tumor tissue samples. Eight normal tissue samples were not detected. Our results suggest that *B. megaterium* presence may play a role in lung cancer carcinogenesis. The tumor microenvironments of *B. megaterium*–positive patients tended to have high levels Th2 cells and low levels of aDC, NKT and plasma cells.

The clinically isolated human pathogen *M. franklinii* is a mycobacterial species and member of the *M. chelonae-M. abscessus* complex. Clinical laboratories typically diagnose *M. franklinii* infections by partially sequencing *rpoB*, *hsp65*, *sodA* and internal transcribed spacer DNA targets, or by complete 16S rRNA gene sequencing analysis in conjunction with assessment of cefoxitin and minocycline susceptibility [42]. Most *M. franklinii* isolates have been collected from the respiratory specimens of patients with underlying pulmonary diseases. In the present study, we detected *M. franklinii* in five lung cancer tissue samples. Eight normal tissue samples were not detected, which may play a role in lung carcinogenesis. Furthermore, high levels of CD4+ T-cells in the tumor microenvironment were associated with *M. franklinii* presences.

The immune responses of cancer patients influence prognosis and survival. In this study, the presence of certain bacteria of lung cancer tissues correlated with poor prognoses and short survival times. Furthermore, the immune responses of presented patients were disrupted, reflected in higher proportions of immune-suppressing cells and decreased TCR and BCR diversity, which may have increased the likelihood of bacterial presence. Our results suggest that presences in cancer tissues likely occurred due to the weak immune systems of the patients, which increased their susceptibility to bacterial presence.

A limitation of this study was the small number of cases, which resulted in a lack of significance for some parameters, and could have led to biased and unclear conclusions. Furthermore, the depth of the RNA-seq metatranscriptomics data was low, which may have resulted in false-negative results. There were also some limitations to the analysis tools applied in this study. However, to overcome these, we performed multiple analyses using different tools to ensure the reliability of the results.

## 5. Conclusions

In this study, we employed RNA-seq to investigate the metatranscriptome of lung cancer patients. Nine bacteria were significantly associated with reduced OS. The presence of two bacterial species, *B. megaterium* and *M. franklinii*, may play an important role in lung tumor carcinogenesis. We also found a correlation between metatranscriptomic changes in expression profiles and tumor immune cell enrichment, as well as immune receptor repertoires.

## Figures and Tables

**Figure 1 genes-12-01458-f001:**
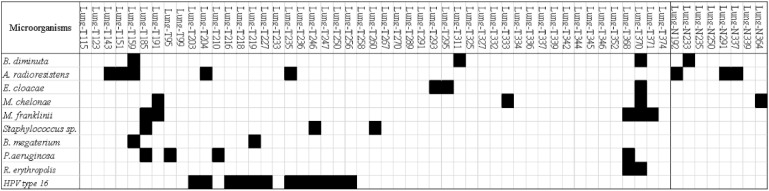
Microorganism composition of the lungs revealed by metatranscriptomic sequencing.

**Figure 2 genes-12-01458-f002:**
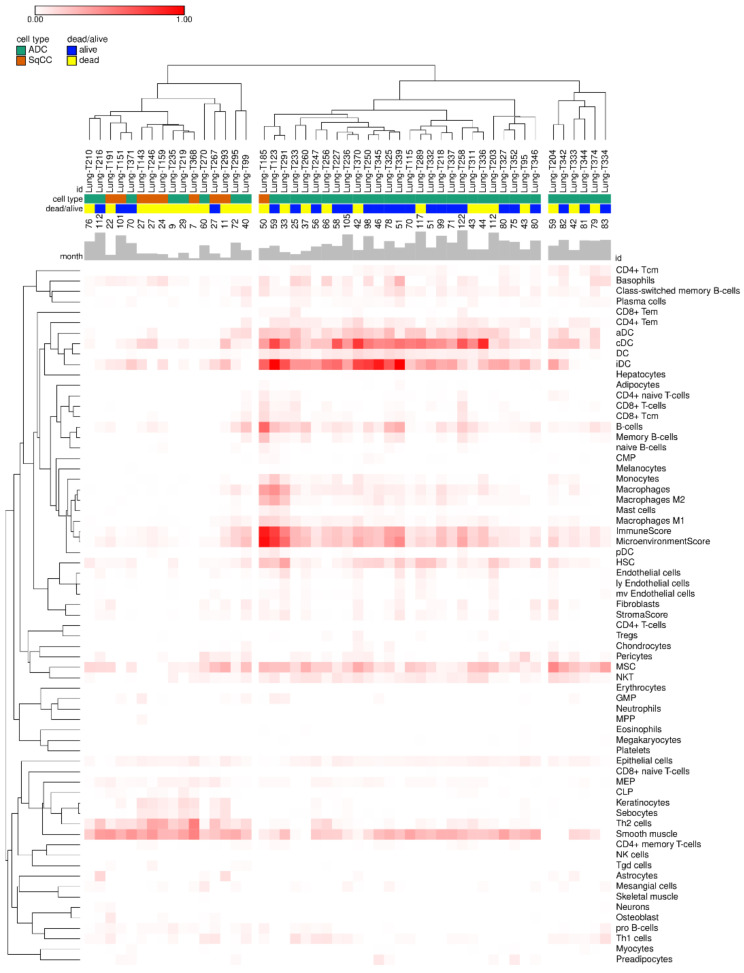
Heatmap showing the 64 cell types found across all samples. Columns are sample id and annotated tissue type with adenocarcinoma (ADC, green) and squamous cell carcinoma (SqCC, orange), survival status with dead (yellow) and alive (blue), survival/follow-up time in month with bar plot. Rows are cell/summary score categories from xCell. Values in heatmap plot are xCell score of each category and samples, and the color map is from white (0) to red (1). Hierarchical clustering was performed with pearson correlation for sample distance calculation and linkage method is “average” on web tool (https://software.broadinstitute.org/morpheus/, accessed on 16 August 2021). The three clusters were generated from manually determined distance cut-off.

**Figure 3 genes-12-01458-f003:**
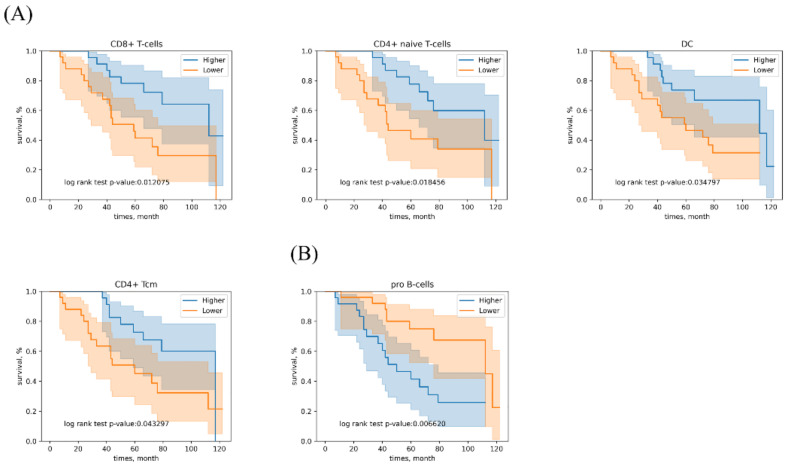
Kaplan-Meier plot showing overall survival (OS) according to immune cell types. (**A**) The group with high levels of CD8+ T-cells, CD4+ naive T-cells, DC and CD4+ Tcm exhibited significantly higher OS according to the log-rank test. (**B**) The group with high levels of pro B-cells showed significantly reduced OS.

**Figure 4 genes-12-01458-f004:**
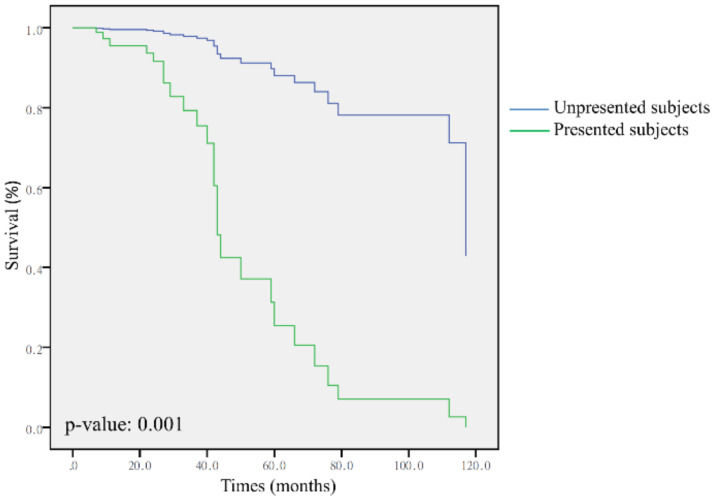
Multivariate survival analysis according to presence or non-presence bacteria in patients with lung cancer.

**Figure 5 genes-12-01458-f005:**
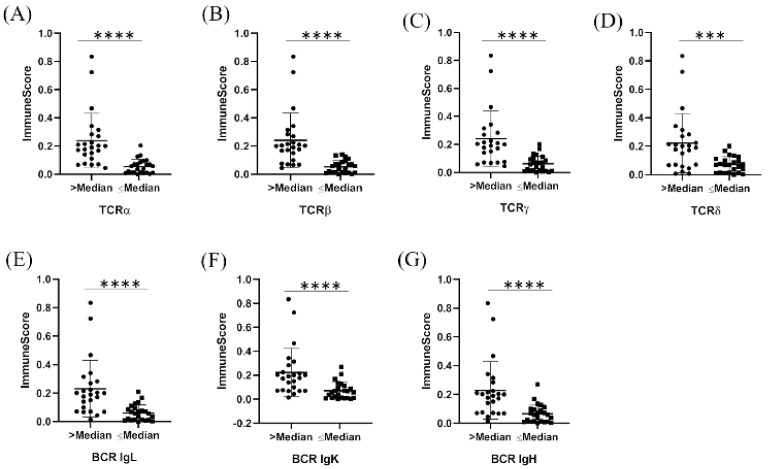
Correlation between clone number of immune receptor repertoires and the Immunoscore, calculated with xCell. (**A**) TCRα, (**B**) TCRβ, (**C**) TCRγ, (**D**)TCRδ, (**E**) BCR IgL, (**F**) BCR IgK, (**G**) BCR IgH. *** *p* < 0.001, **** *p* < 0.0001.

**Figure 6 genes-12-01458-f006:**
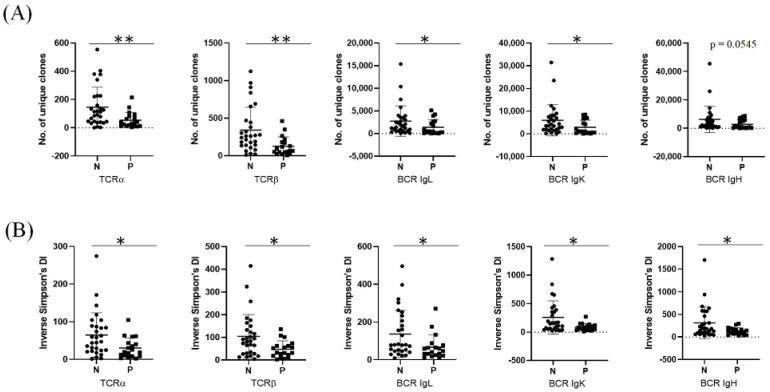
Comparison of the immune receptor repertoires between presented and unpresented subjects. (**A**) Number of unique clone and (**B**) clonal diversity. * *p* < 0.05, ** *p* < 0.01.

**Table 1 genes-12-01458-t001:** Multivariate analysis (Cox regression) of independent prognostic factors in patients with lung cancer.

Variables		Hazard Ratio	95% CI	*p*-Value
TCRα clone number	≤Median	1	0.101–0.972	0.044
	>Median	0.314		
Stage	I	1		0.023
	II	0.844	0.242–2.946	0.791
	III	2.471	0.612–9.969	0.204
	IV	59.625	4.009–886.782	0.003
Grade	1	1		0.523
	2	0.512	0.044–5.930	0.592
	3	0.334	0.022–5.070	0.429
	4	3.269	0.056–191.120	0.568
Gender	M	1	0.578–6.2	0.289
	F	1.907		
Histologic cell type	ADC	1	8.743–513.175	0.000
	SqCC	66.984		

## Data Availability

The RNA-Seq data from this study was submitted to the NCBI Sequence read Archive (SRA) under BioProject accession nos. PRJNA698419.

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
