# Peer review of "Metatranscriptomic Analysis of Human Lung Metagenomes from Patients with Lung Cancer"

_genes, 2021, doi:10.3390/genes12091458_

Round 1

Reviewer 1 Report

The authors had performed RNA-seq analysis to investigate the metatranscriptome of lung cancer patients and have clearly presented their findings. Though no direct causative factors have been identified, a clear correlation with two bacterial species, B. megaterium and M. franklinii have been identified. 

Overall, the results were clearly stated and the work is of significance to the field of study. Good job!

Author Response

Thank you for your comment.

Reviewer 2 Report

Reviewer’s comment to the author:

In the manuscript entitled “Metatranscriptomic analysis of human lung metagenomes from patients with lung cancer” authors have studied the microbiome in the lung tumors and identified a panel of 9 bacteria showing correlation with overall survival of patients.  In this study, Authors have also investigated the correlation of Tumor Immune Cells enrichment with the disease survival and the presence of the bacteria. Though the findings of this manuscript are interesting; however, there are other serious concerns that warrants this manuscript unsuitable for the publication in its present form.

  1. First and foremost, the findings of this paper are not enough to be published as an original article considering the small sample size and insufficient number of patients.
  2. Many of bacterial species which show prognostic significance in log rank test are present in very less percentage (2-8%) of tumors. A larger sample size should be included to draw an un-biased and clear conclusion. The bacterial species with higher incidence shows not a very strong (p values closer to 0.05) association with survival.  So, Increasing the sample size is important in order to get a clear picture and to improve on the reliability of results.
  3. Authors have used SAMSA2 and Kraken2 tools to detect bacteria in met transcriptome data from lung tumor samples, aeruginosa is present in 0 sample as shown in supplementary table S1 (table S1 is list of bacterial species detected by either tools, but as per supplementary table S2, it is present in 4 samples (table S2 is list of bacterial species detected by both the tools). I would suggest the authors to relook at these figures. P. aeruginosa is one of 9 bacterial species which is reported as showing significant correlation with survival. So, it becomes imperative to relook at these analysis results.
  4. Quality/resolution of Fig. 3 Kaplan- Meier Plot is not satisfactory as the p values are not clearly visible on any of plots even after magnifying it.
  5. Fig.3A shows high levels of CD8+ T-cells, CD4+ naive T-cells, DC and CD4+ Tcm exhibited significantly higher OS whereas Fig. 4 shows that lower abundances of CD8+ T-cells, CD4+ naive T-cells and DC in presented subjects were predictive of favorable outcomes. These are two contradictory findings/observations which should be discussed in detail and explained by the Authors. Although discussion is well written, it did not touch upon this aspect.
  6. Fig.4 is not interpreted clearly. The Fig. 4 shows that the immune score for CD4+ T cells, CD8+ T cells and DC cells are high in unpresented subjects as compared to presented subjects which indicate towards the suppression of these immune cells by presence of bacteria.
  7. Supplementary table S4 shows the correlation between levels of tumor immune cells with the presence of bacterial species. Some bacteria correlate positively with levels of tumor immune cells while some correlate negatively. Author should analyze whether the positive effect of presence of bacteria on the survival is related to their positive effect on levels of tumor immune cells or it is an independent effect. The author should also evaluate the utility of immune tumor cells and the presence of bacteria in combination as a prognostic factor. 
  8. Authors have shown positive correlation of levels of tumor immune cells with disease free survival. They should also do Multivariate analysis in order to find out how it performs as an independent prognostic factor.

please also find the attached word document for the same comments.

Thank you!

Round 2

Reviewer 2 Report

Authors have addressed all my comments raised in my previous review except for my concern of the small sample size and modified the paper accordingly, and the paper now looks more acceptable. Though I still feel that there should be a larger sample size for this kind of studies to draw an un-biased and clear conclusion.

One minor correction is required, on page 07, In the line 233, Author have mentioned that The B. diminuta, A. radioresistens, M. chelonae and M. franklinii maintained the significance after adjustment for tumor immune cells (p=0.031, 0.070, 0.006 and 0.031, respectively). For A. radioresistens, p value is 0.070 which is not significant. Author should correct this statement.

Thank you

Author Response

Reviewer #2:

Authors have addressed all my comments raised in my previous review except for my concern of the small sample size and modified the paper accordingly, and the paper now looks more acceptable. Though I still feel that there should be a larger sample size for this kind of studies to draw an un-biased and clear conclusion.
Response: Thank you for your comment. We have added the following sentence to the Discussion: A limitation of this study was the small number of cases, which resulted in a lack of significance for some parameters, draw biased and unclear conclusion.  (Page 9, line 316) 

One minor correction is required, on page 07, In the line 233, Author have mentioned that The B. diminuta, A. radioresistens, M. chelonae and M. franklinii maintained the significance after adjustment for tumor immune cells (p=0.031, 0.070, 0.006 and 0.031, respectively). For A. radioresistens, p value is 0.070 which is not significant. Author should correct this statement."

Response: The sentence has been rewritten: The B. diminuta, M. chelonae and M. franklinii maintained the significance after adjustment for tumor immune cells (p=0.031, 0.006 and 0.031, respectively) (Table S6). After adjustment, the A. radioresistens, E. cloacae, Staphylococcus sp., B. megaterium and R. erythropolis showed no significance of the relationship.  (Page 7, line 233)